# Long-Chain Polyunsaturated Fatty Acids, Homocysteine at Birth and Fatty Acid Desaturase Gene Cluster Polymorphisms Are Associated with Children’s Processing Speed up to Age 9 Years

**DOI:** 10.3390/nu13010131

**Published:** 2020-12-31

**Authors:** Cristina Campoy, Hatim Azaryah, Francisco J. Torres-Espínola, Cristina Martínez-Zaldívar, José Antonio García-Santos, Hans Demmelmair, Gudrun Haile, Peter Rzehak, Berthold Koletzko, Eszter Györei, Tamas Décsi, María del Carmen Ramírez-Tortosa, Eva Reischl, Anne M. Molloy, Juan de Dios Luna, Miguel Pérez-García

**Affiliations:** 1EURISTIKOS Excellence Centre for Paediatric Research, Biomedical Research Centre, University of Granada, 180016 Granada, Spain; rifappstudio@gmail.com (H.A.); fjtespinola@yahoo.es (F.J.T.-E.); mcriszald@ugr.es (C.M.-Z.); joseantonio_gsantos@outlook.es (J.A.G.-S.); 2Department of Paediatrics, School of Medicine, University of Granada, Avda, Investigación 11, 180016 Granada, Spain; 3Spanish Network of Biomedical Research in Epidemiology and Public Health (CIBERESP), Granada’s Node, Institute of Health Carlos III, 28029 Madrid, Spain; 4Instituto de Investigación Biosanitaria de Granada (Ibs-GRANADA), Health Sciences Technological Park, 18012 Granada, Spain; 5Instituto de Neurociencias “Doctor Olóriz”, Health Sciences Technological Park, 18012 Granada, Spain; 6Ludwig-Maximilians-Universität München, Department of Paediatrics, Paediatrics, Dr. von Hauner Children’s Hospital, University of Munich Hospitals, 80337 Munich, Germany; hans.demmelmair@med.uni-muenchen.de (H.D.); gudrun.haile@med.uni-muenchen.de (G.H.); peter.rzehak@med.uni-muenchen.de (P.R.); berthold.koletzko@med.uni-muenchen.de (B.K.); 7Department of Paediatrics, University of Pécs, 7623 Pécs, József Attila u. 7, 7623 Pécs, Hungary; gyorei.eszter@pte.hu (E.G.); decsi.tamas@pte.hu (T.D.); 8Department of Biochemistry and Molecular Biology II, Institute of Nutrition and Food Technology “José Mataix Verdú”, Biomedical Research Center, University of Granada, 18100 Granada, Spain; mramirez@ugr.es; 9Helmholtz Zentrum Munchen, Research Unit of Molecular Epidemiology, D-85764 Neuherberg, Germany; eva.reischl@helmholtz-muenchen.de; 10School of Medicine, Trinity College, 152–160 Pearse Street, D02 Dublin 2, Ireland; amolloy@tcd.ie; 11Department of Biostatistics, School of Medicine, University of Granada, 18016 Granada, Spain; jdluna@ugr.es; 12Mind, Brain and Behaviour International Research Centre (CIMCYC), University of Granada, 18011 Granada, Spain; mperezg@ugr.es; 13Spanish Network of Biomedical Research Centre on Mental Health (CIBERSAM), Granada’s Node, Institute of Health Carlos III, 28029 Madrid, Spain

**Keywords:** long-chain polyunsaturated fatty acids, folate, prenatal supplementation, processing speed, neurodevelopment, *FADS* gene, children

## Abstract

Both pre- and early postnatal supplementation with docosahexaenoic acid (DHA), arachidonic acid (AA) and folate have been related to neural development, but their long-term effects on later neural function remain unclear. We evaluated the long-term effects of maternal prenatal supplementation with fish-oil (FO), 5-methyltetrahydrofolate (5-MTHF), placebo or FO + 5-MTHF, as well as the role of fatty acid desaturase (*FADS*) gene cluster polymorphisms, on their offspring’s processing speed at later school age. This study was conducted in NUHEAL children at 7.5 (*n* = 143) and 9 years of age (*n* = 127). Processing speed tasks were assessed using Symbol Digit Modalities Test (SDMT), Children Color Trails Test (CCTT) and Stroop Color and Word Test (SCWT). Long-chain polyunsaturated fatty acids, folate and total homocysteine (tHcy) levels were determined at delivery from maternal and cord blood samples. *FADS* and methylenetetrahydrofolate reductase (*MTHFR*) 677 C > T genetic polymorphisms were analyzed. Mixed models (linear and logistic) were performed. There were significant differences in processing speed performance among children at different ages (*p* < 0.001). The type of prenatal supplementation had no effect on processing speed in children up to 9 years. Secondary exploratory analyses indicated that children born to mothers with higher AA/DHA ratio at delivery (*p* < 0.001) and heterozygotes for *FADS1* rs174556 (*p* < 0.05) showed better performance in processing speed at 9 years. Negative associations between processing speed scores and maternal tHcy levels at delivery were found. Our findings suggest speed processing development in children up to 9 years could be related to maternal factors, including AA/DHA and tHcy levels, and their genetic background, mainly *FADS* polymorphism. These considerations support that maternal prenatal supplementation should be quantitatively adequate and individualized to obtain better brain development and mental performance in the offspring.

## 1. Introduction

The speed of information processing is essential for higher order cognitive functions, including memory or executive functions [1]. Processing speed can be defined as the time required to move information from one neuron to the next [2], or how quickly a person can perform the mental operations needed to complete a task [3]. This cognitive process is highly related to intact myelination, which is important for the integration of information across spatially distributed neural networks. Moreover, the association between white matter integrity and processing speed in cognitive tasks has been consistently established [4,5].

Long-chain polyunsaturated fatty acids (LC-PUFAs) and folic acid play an important role in brain development, particularly during fetal and early postnatal life [6,7]. Interestingly, their effects on neurodevelopment depend on timing of occurrence and brain needs for particular nutrients at that time [8]. For instance, essential fatty acids (FA) deficiencies during the first year of life lead to severe impairments in synapse formation and myelination [9], which may have negative effects on processing speed tasks later in life [10].

LC-PUFAs, particularly docosahexaenoic acid (DHA, 22:6 *n* − 3) and arachidonic acid (AA, 20:4 *n* − 6) are incorporated into the brain in relatively large amounts during the pre- and postnatal growth spurt [11,12,13]. However, due to a limited capacity in the fetus and neonate for PUFAs elongation and desaturation, tissue deposition of DHA and AA strongly depends on the pre-formed LC-PUFAs supply via the placenta and postnatal diet [14]. The availability of different PUFAs is also dependent on genetic polymorphisms in the fatty acid desaturase (*FADS*) gene cluster [15]. There is evidence that dietary LC-PUFAs supply in early life may modulate information processing [12,16], cognitive and visual development [17,18], as well as early mental and motor skills development [6,19]. Recently, higher maternal DHA status has been also related to better performance in language and short-term memory in the offspring [20]. Moreover, early availability of *n −* 6 PUFAs, mainly AA, during pre- and postnatal periods has been positively associated to cognitive performance and mental function in later childhood [21,22,23].

Folate intake before and during pregnancy is also essential for normal brain development, differentiation and cognitive performance [24,25,26,27]. Maternal folate deficiency causes structural brain abnormalities during fetal development and poor childhood cognitive ability [26], while maternal folate supplementation in pregnancy improves neurological development and may reduce the prevalence of autism spectrum disorders in their offspring [24,28].

Having in mind that LC-PUFAs and folate play key roles in synaptogenesis, synapse maturation and myelination, dietary intake of both nutrients might have effects on processing speed [2]. Long-term effects of nutritional interventions on processing speed have been classically evaluated using perceptual speed tasks [29], but further studies evaluating specific neuropsychological domains are still missing [6,30]. Therefore, our main objective was to evaluate the potential role of LC-PUFAs and total homocysteine (functional maker of folate status) at delivery, as well as maternal *FADS* polymorphisms, on children´s processing speed at 7.5 and 9 years.

## 2. Materials and Methods

### 2.1. Study Design and Subjects

This is a follow-up study of the NUHEAL (Nutraceuticals for a Healthier Life) trial, registered at www.ClinicalTrials.gov, Identifier NCT01180933. Detailed study design, subject recruitment, and population characteristics have been described elsewhere [31,32]. Briefly, NUHEAL project is a multicenter, randomized, double-blind, placebo-controlled trial in healthy pregnant women from Munich (Germany), Pécs (Hungary) and Granada (Spain). Children from Hungary were not included in the present analysis due to high number of missing values. Pregnant women were assigned by blockwise randomization to receive either a modified fish-oil (FO) preparation [500 mg DHA + 150 mg eicosapentaenoic acid (EPA)/day], 5-methyl-tetrahydrofolate (5-MTHF) (400 µg/day), a combination of both supplements (FO + 5-MTHF), or placebo, from gestational week 20 until delivery. Detailed information on sociodemographic data and course of pregnancy together with maternal blood samples were collected at 20 and 30 weeks of pregnancy and at delivery; additionally, cord blood samples were also obtained.

Within NUTRIMENTHE EU Project (grant agreement no. 212652) framework, of 270 women participating in NUHEAL study until giving birth, 152 mothers agreed to participate in the follow-up for their offspring at 7.5 and 9 years of age. Processing speed tasks were entirely performed in 143 NUHEAL children at 7.5 years (FO, *n* = 38; 5-MTHF, *n* = 29; FO + 5-MTHF, *n* = 36; placebo *n* = 40) and 127 NUHEAL children at 9 years (FO, *n* = 34; 5-MTHF, *n* = 28; FO + 5-MTHF = 28; placebo, *n* = 37) (Figure 1).

The follow-up study protocols were approved by the Ethical Committees from all centers involved in the study. Written informed consent was obtained from parents of all participating children at study entry and at each follow-up.

### 2.2. Neuropsychological Assessment

NUTRIMENTHE Neuropsychological Battery (NNB) was used to evaluate the whole spectrum of neuropsychological functioning in children aged 7.5 and 9 years, including processing speed tasks such as Symbol Digit Modalities Test (SDMT), Children Color Trails Test (part 1) (CCTT-1) and Stroop Color and Word Test (SCWT) [33].

The Symbol Digit Modalities Test (SDMT) was used to assess information-processing speed and attention [34]. This test requires individuals to write the correct number under the corresponding symbol according to a key code specified on the top of the page, which links different meaningless geometric symbols with numbers 1 through 9. The participant is given 90 s to complete the task. The number of correctly identified symbols (hits) is recorded as a score, interpreting higher scores to be an indicator of better child processing speed.

The Children Color Trails Test (CCTT) is an individually administered neuropsychological instrument which consists of two parts used to evaluate sustained visual attention, sequencing, psychomotor speed (part-1), and cognitive flexibility (part-2). This test requires the connection of one set of encircled numbers (1–25) in ascending order. Current study used only the part 1 of this test, which even numbers are printed in a yellow background while odd numbers are printed in a pink background. The final score is the time (in seconds) taken to complete part-1 of the CCTT [35], so that shortest time in this test is related to better processing speed of participant.

Finally, Golden’s version of the Stroop Color and Word Test (SCWT) was used to evaluate selective attention, cognitive inhibition and information processing speed [36]. There are three components to this task. First, participants are asked to read aloud color words (blue, green and red) printed in black ink. Second, the child is asked to say the colors of “XXXX” printed in blue, green or red. Finally, the child is asked to name the ink color of color words (blue, green or red) printed in incongruent colors as quickly and accurately as possible in 45 s time. As consequence, this test produces three direct scores: the word-reading score, the color-naming score, and the color-word score, respectively [37]. The increase in time taken to perform the color-word test compared with the word-reading and color-naming tests is called color-word interference effect or Stroop effect, which is considered as the main dependent variable for SCWT test [38]. Both word-reading and color-naming scores (hits) are related to processing speed of congruent semantic information (high scores represent better processing speed), but color-word score is associated to attention development and, for this reason, it has not been taken into account in the current study.

### 2.3. Fatty Acid Analyses in Maternal and Umbilical Cord Plasma Phospholipids

Procedures of analysis for FA determinations in plasma phospholipids have been described in detail elsewhere [31,32]. Briefly, blood was centrifuged at 3500× *g* for 10 min at room temperature within 2 h. Plasma was thereafter removed and stored at −80 °C. Lipid extraction from plasma was performed according to the method of Kolarovic and Fournier [39]. Analysis of FA methyl esters from plasma phospholipids was performed by high-resolution capillary gas-liquid chromatography. Conditions during the analysis and standards used were described elsewhere [40]. Results were expressed as weight percentages (wt %) of all quantified FA.

### 2.4. Folate Analysis

Analysis of plasma folate was carried out by microbiological assay using a chloramphenicol resistant strain of *Lactobacillus casei*, as previously described [41]. Inter and intra-assay coefficients of variation were below of 11%.

### 2.5. Total Homocysteine

Total homocysteine (tHcy) concentrations were assayed by fluorescence polarized immunoassay on the IMx autoanalyser [42]. Sample preparation and chromatographic conditions were performance as described previously [43]. The fluorescence intensities were measured with excitation at 385 nm and emission at 515 nm.

### 2.6. SNP Selection and Genotyping

Genotyping of single-nucleotide polymorphisms (SNPs) from the *FADS* gene cluster was performed with the iPLEX method (Sequenom, San Diego, CA, USA) by means of matrix-assisted laser desorption ionization-time of flight mass spectrometry method (MALDI-TOF MS, Mass Array, Sequenom), according to the manufacturer´s instructions, as previously described [44]. Standard genotyping quality control included 10% duplicate and negative samples. Genotyping discordance rate was below 0.3%. SNP selection for following analyses is showed in Appendix A.

### 2.7. MTHFR 677 C/T Polymorphism

Genomic DNA was prepared from maternal and umbilical cord blood samples obtained at delivery. DNA samples were genotyped for the methylenetetrahydrofolate reductase (MTHFR) 677 C/T variant by polymerase chain reaction (PCR), restriction enzyme digestion, and DNA fragment separation by electrophoresis, as described previously [42]. MTHFR 677 C > T was selected for its high clinical relevance in humans due to the association of TT genotype with high plasma homocysteine concentrations [45].

### 2.8. Statistical Analysis

Using standard approaches, statistical power for the current study was calculated, setting α value as 0.05 and β value as 0.2. Different intervention groups (FO, 5-MTHF and FO + 5-MTHF) were grouped and compared with placebo group for processing speed tasks, including SDMT, CCTT and SCWT, in children at 7.5 years old. A statistical power of 80% was obtained for selected tasks in aforementioned population (Appendix A). “PowerEQTL v0.1.3” (R software) was also used to calculate the statistical power for our genetic study; except for FADS2 rs174570 and FADS3 rs2727271, a statistical power of 90% was obtained for analyzed SNPs (Appendix A).

A descriptive analysis of quantitative variables was performed using summary measures (mean, standard deviation (SD), standard error of the mean, centiles, median, interquartile or amplitude ranges), meanwhile frequency distribution was used for qualitative variables. Comparisons among different groups of treatment (FO, 5-MTHF, both or placebo) were made using one-way ANOVA for continuous variables or χ^2^ test for categorical variables. In order to verify the underlying hypothesis of one-way ANOVA (variance’s homogeneity and normality), Box-Cox transformation was computed when considered necessary. When one-way ANOVA resulted significant, Bonferroni test post-hoc was applied.

Variables of processing speed obtained from SDMT, CCTT-1 and SCWT tests were considered as dependent variables. Considering the hierarchical structure resulted from processing speed evaluation in children at 7.5 and 9 years, mixed models for repeated measures were performed considering as fixed effects a set of potential confounding variables, such as study group, time point, country of origin, maternal age, hematocrit levels, parity, gravidity risk maternal education level, maternal smoking, maternal BMI, mode of delivery, gestational age, child’s sex and mother’s biochemical and molecular parameters at delivery (AA/DHA ratio, FADS1 rs174556 polymorphism, total homocysteine, MTHFR C677T polymorphisms, plasma folate). Interactions between study group and time point were also studied. The random effects were the subjects nested ID to estimate the intra variance. Logistic regression mixed model was used when dependent variables were dichotomized (below or above percentile 25, 50 and 75), in which case the measure of the effect was the suitable odds ratio with corresponding confidence interval (Appendix A). All potential confounders were added at once and selected those which significance *p* < 0.05.

All statistical analyses were performed using the statistical package STATA 12.1 (Stata Corp, College Station, TX, USA). *p* values < 0.05 were considered as statistically significant.

## 3. Results

### 3.1. Background and Baseline Characteristics of the NUHEAL Study Participants

The baseline characteristics of the mothers whose children were evaluated at 7.5 and 9 years of age are shown in Table 1. No difference between prenatal supplementation groups were observed in those descriptive variables analyzed prior or during pregnancy, including country of origin, maternal age, BMI at 20 and 30 weeks of pregnancy, mode of delivery, hematocrit at 30 weeks of pregnancy, parity, smoking during pregnancy, gravidity risk at 20 weeks of pregnancy, high maternal education, family status or gestational age at delivery. Moreover, we analyzed maternal biochemical parameters at delivery. As expected, plasma levels of folate were significantly higher in those mothers who received 5-MTHF or FO + 5-MTHF supplementation during pregnancy (*p* < 0.001). However, type of prenatal supplementation had no effect on maternal AA/DHA ratio and tHcy levels.

### 3.2. Processing Speed Task of the NUHEAL Children at 7.5 and 9 Years Old

As shown in Table 2, no significant statistical differences were found among the type of prenatal supplementation in all analyzed processing speed tasks at 7.5 years, except for CCTT-1 test. In fact, children born to mothers who were supplemented with 5-MTHF during pregnancy showed a decrease in the timing to solve CCTT-1 (*p* = 0.017). Moreover, there were no differences between type of prenatal supplementation and processing speed tasks at 9 years old. In general, we observed that children aged 9 years had better processing speed than those aged 7.5 years in terms of higher scores and less time to solve CCTT-1 test.

### 3.3. Prenatal Predictors of Processing Speed in Children up to 9 Years

#### 3.3.1. Symbol Digit Modalities Test (SDMT) Hits

Type of prenatal supplementation had no long-term effects on SDMT hits in children at 7.5 and 9 years, (Table 3). However, we observed an age-dependent increase in SDMT hits; in fact, children at 9 years showed an increase of 10.72 points in the mean of hits [(95% CI: 7.98–13.47); *p* < 0.001] compared to children at 7.5 years. Moreover, each unit of increase in maternal AA/DHA ratio at delivery predicted an increase of 8.80 points in the mean of SDMT hits [(95% CI: 3.61–13.99); *p* = 0.001]. Conversely, higher maternal weight gain between 20 and 30 weeks of pregnancy (dBMI) [(95% CI: −1.25–−0.02); *p* = 0.043] and maternal tHcy [(95% CI: −1.02–−0.002); *p* = 0.049] were associated with a decrease of 0.63 and 0.52 points in the mean of SDMT hits, respectively.

Further logistic regression analysis, characterizing those children aged 9 years having hits above P75, showed that maternal AA/DHA ratio at delivery was the best predictor to obtain higher number of ***SMDT*** hits [OR: 30.46 (95% CI: 3.68–252.0); *p* = 0.002] (Table 4). This probability is also increased in those children whose mothers had a high educational level (*p* = 0.013) or were heterozygote for *FADS1* rs174556 (*p* = 0.014). However, maternal tHcy at delivery reduced the odds of placing children aged 9 years above the P75, with a probability of 0.71 to obtain less hits per each unit (μmol/L) of increase of tHcy [OR: 0.71 (95% CI: 0.56–0.89); *p* = 0.003].

#### 3.3.2. Children Color Trails Test (CCTT−1)

No statistically significant differences in elapsed time for ***CCTT-1*** at 7.5 and 9 years were found among groups of prenatal supplementation after adjustment for selected confounders. Independently of prenatal supplementation, children aged 9 years showed a decrease in the timing spent to solve the task compared to their results at the previous examination at age 7.5 years (*p* < 0.001) (Table 3).

After considering the interaction between age and prenatal supplementation, we observed that decrease in time elapsed was higher in those children whose mothers were supplemented with FO during pregnancy (*p* = 0.0001), 5-MTHF (*p* = 0.0113) or FO + 5-MTHF (*p* = 0.038) (Figure 2), but not in the placebo group.

As shown in Table 3, other selected cofounders had also a significant association on the time required to complete CCTT-1 test. In fact, vacuum delivery determined an increase in the time elapsed of 24.77 s for the task (*p* = 0.010) compared to those children whose mothers had an uncomplicated spontaneous delivery. Moreover, smoking during pregnancy had also a negative effect on the child’s CCTT-1 hits, increasing the time of solving this task in 23.43 s (*p* = 0.001). Finally, we observed that maternal AA/DHA ratio at delivery was a significant factor determining less timing to solve the CCTT-1 task of their offspring up to 9 years (−27.87 s, *p* = 0.019). Positive association between maternal AA/DHA ratio and CCTT-1 was also determined after logistic regression analysis, and characterizing those children aged 9 years with elapsed time of solving the CCTT-1 below the P25 [OR: 18.53 (95% CI: 2.13–160.9); *p* = 0.008] (Table 4). Shorter solving times were also observed in those children born to mothers who were heterozygote for FADS1 rs174556 [OR: 2.75 (95% CI: 1.02–7.39; *p* = 0.045].

#### 3.3.3. Stroop Color and Word Test (SCWT)

We analyzed the influence of prenatal supplementation and other selected cofounders on the information processing speed using Stroop Color and Word test and its obtained scores: Word-Reading (WRST) and Color-Naming (CNST).

Similarly to initial descriptive evaluation, further adjusted analysis for selected cofounders did not show statistically significant differences between different types of prenatal supplementation in the number of WRST hits obtained by children at 7.5 and 9 years (Table 3). However, independently of maternal supplementation during pregnancy, children at 9 years showed an increase of 15.37 points in the number of hits for WRST (*p* < 0.001) respect to children at 7.5 years. In relation to other cofounders considered in the model, boys showed 4.93 less hits than girls solving the task (*p* = 0.019). Moreover, we also observed that each increment of one unit (µmol/L) of maternal tHcy at delivery predicted that their offspring had 1.09 less hits by average solving the task (*p* = 0.025). Further logistic regression analysis, characterizing those children aged 9 years which hits of solving the WRST test below the P50, determined that maternal tHcy at delivery had a negative influence on this test [OR: 0.69 (95% CI: 0.51–0.93); *p* = 0.015] (Table 4).

When analyzing the effects of both prenatal supplementation and child´s age on CNST, similar results to those reported above were found. In fact, cofounder adjustment analysis did not show statistical differences between different groups of prenatal supplementation in the number of hits obtained by NUHEAL children at 7.5 and 9 years. Again, there was an increase of CNST hits at 9 years compared to 7.5 years (*p* < 0.001), but this effect was independent of type of supplementation during pregnancy (Table 3). Interestingly, we observed that each increment of one unit of AA/DHA ratio in maternal blood at delivery determined that their offspring had 6.55 more hits by average solving the *CNST* (*p* = 0.034) (Table 3). After logistic regression analysis, we observed that maternal age positively influenced the likelihood for children aged 9 years to be above the P75 to solve *CNST* test (OR: 1.20 (95% CI: 1.01–1.42); *p =* 0.039), but maternal tHcy level at delivery reduced this probability (OR: 0.51 (95% CI: 0.32–0.81); *p* = 0.005) (Table 4).

## 4. Discussion

This study was performed to analyze the long-term effects of prenatal supplementation, as well as maternal FADS and MTHFR genetic polymorphisms, on processing speed in healthy school-age children. Our results suggest that neither FO nor folate prenatal supplementation predicted high processing speed scores at school-age children. However, our secondary exploratory analyses seem to indicate that maternal AA/DHA ratio and FADS1 rs174556 SNPs, were positively associated with later processing speed in their offspring up to 9 years, particularly in SDMT and CCTT-1 tests, while tHcy concentrations in maternal plasma at delivery showed a negatively effect on child processing speed, according to results obtained in SDMT, WRST and CNST tests.

Prenatal folic acid supplementation has been related to better neurodevelopment in offspring, in terms of reducing the risk of behavioral problems [46], language delay [47], inattention [24], hyperactivity and peer problems [4,48,49]. However, their potential effects on cognitive and mental performance during development are inconsistent, partly due to the very limited number of studies published [50]. Folic acid acts as methyl donor in the metabolic conversion of homocysteine (Hcy) to methionine [51]. As consequence, poor folate status, by itself or in combination with a poor status of other B-vitamins, attenuates this metabolic pathway, which subsequently increases total Hcy levels. There is evidence that higher maternal Hcy (≥8.3 µmol/L) may not only negatively influence placental development, birth weight and pregnancy outcomes [52], but is also related to cytotoxic- and oxidative stress-dependent endothelial cell impairment and apoptosis of placental trophoblast [53,54]. Our results suggest a negative relation between maternal tHcy and the child´s later cognitive function. Children born to mothers with high tHcy levels during pregnancy showed a decrease in SDMT and SCWT hits, as well as lower likelihood to be in the upper percentiles of the WRST and CNST. Because tHcy concentrations are a functional indicator of folate status, folate supplementation before and during pregnancy may enhance neurodevelopment, while early supplementation also reduces the risk of congenital malformations [55].

Fetal blood levels of LC-PUFAs, including DHA and AA levels, are closely related to maternal LC-PUFAs status during pregnancy [32], and play a major role for an optimal brain development [20,56]. DHA is related to synaptogenesis, nerve growth factor expression and neuronal differentiation [57]. AA is involved in several synaptic signaling pathways [58], synthesis of eicosanoids, prostaglandins and leukotrienes [11,12], growth-related early gene expression and cell growth [59,60]. However, controversial results have been reported regarding long-term effects of *n* − 3 or *n* − 6 LC-PUFAs supplementation during pregnancy or lactation on the child´s later neurodevelopment [6,17,31,61,62,63]. As a consequence, there is growing interest to analyze the long-term effects of AA/DHA ratio on child neurodevelopment, which reflects both its endogenous synthesis and exogenous supply. Moreover, because both FAs compete for the same enzymatic pathways to convert them into biologically active eicosanoids, AA/DHA ratio is strongly influenced by the prevalence of genetic predisposition for *FADS* and elongase genes [57]. Higher DHA/AA ratio and higher DHA concentrations in cord blood have been considered beneficial for infant visual, cognitive and motor development in Arctic Inuit exposed to high intakes of seafood and *n −* 3 LC-PUFAs [64]. In our study, we did not find clear associations between type of prenatal supplementation and processing speed development, except for beneficial effects of prenatal FO supplementation on CCTT-1 elapsed time at 9 years. However, a higher maternal AA/DHA ratio seems to be a positive and long-term modulator of processing speed (mainly on SDMT, CCTT-1 and CNST) in the offspring, indicating the importance not only of DHA but also its equilibrium with AA. Interestingly, at 9 years, children whose mothers were heterozygotes for FADS1 rs174556 performed better the processing speed tasks respect to those born to mothers with homozygous major alleles. Our findings are also consistent with a role of PUFAs in myelination and white matter integrity, as shown in animal studies [65,66,67]. In this regard, DHA may increase processing speed by changing the physical-chemical and structural properties of membrane [68]. Moreover, Peters et al. [69] not only demonstrated that erythrocyte membrane PUFAs concentrations in young adults seem to be robustly related to white matter integrity, but also showed that these associations were mostly related to AA levels. Since a connection between white matter integrity and processing speed in cognitive tasks has been established, our results seem to show positive and strong long-term effects of perinatal LC-PUFAs, in terms of adequate AA/DHA ratio and *FADS1* polymorphism, on cognitive development, suggesting an increase of white matter volume and better integrity.

The major strength of the present study is the long-term follow-up, from pregnancy to age 9 years, allowing us to obtain evaluation of long-term effects of prenatal supplementation with FO, 5-MTHF or FO + 5-MTHF on the child’s cognitive abilities. Moreover, the NUHEAL study was conducted in three different countries (Spain and Germany) with distinct eating habits. Given that country of origin has been accounted for as a confounder, our data show that long-term effects observed on processing speed are independent of the women’s diet. In this regard, we highlight the influence of the cultural level of the mothers on the processing speed of their children, which increases the need to take into account the different socio-environmental factors during early life that may influence on later speed processing capacities.

Our results have some limitations. The number of children who belong to each study group was homogeneous but relatively low. However, after combining the data from children born to mothers who were supplemented during pregnancy (*n* = 103), the effects of supplement combinations were significant with respect to children whose mothers received placebo. Secondly, the effects of FO, 5-MTHF or FO + 5-MTHF supplementation were not evaluated at different time-points of administration. Moreover, our study has been conducted only in a selected age range (7.5–9 years). Thus, future studies will be necessary to evaluate whether our findings can be extended to other times of administration or at different ages during development. Moreover, conversion of homocysteine to methionine is largely based on both folate and vitamin B12 levels, which act as substrate and cofactor, respectively. However, the status of vitamin B12 level in our study is largely unknown. Finally, neuropsychological tests used to evaluate processing speed were administered by different technicians in each country, although all of them received a common training to reduce the examiner and cultural influences on the results.

Findings obtained in this study should be interpreted with caution. NUHEAL women belonging to FO and FO + 5-MTHF groups received fish oil preparation, alone or in combination with folate, at a dose of 500 mg DHA + 150 mg EPA from week 20 until delivery, which are higher than current international recommendations (200–300 mg of DHA/day) [12,70]. In addition, study participants followed their usual eating patterns, including PUFAs rich food. Therefore, it is of utmost importance to determine the timing, necessary duration and dosage of DHA + EPA supplementation (in equilibrium to AA) during pregnancy, for obtaining the best cognitive development in the offspring. Interestingly, the mixed supplementation including FO and 5-MTHF had no effect on processing speed up to age 9 years. Thus, we propose that maternal supplementation based on folate, DHA and EPA should be individualized, taking into account diet, habits, folate status and maternal *FADS1* genetic variant rs174556 G/A, and perhaps not together at the same time during pregnancy.

## 5. Conclusions

In summary, in our population the maternal AA/DHA ratio at delivery and maternal heterozygosity for the FADS1 genetic variant rs174556 had positive long-term effects on processing speed in the offspring up to 9 years. Processing speed tasks, in terms of less time to solve CCTT-1 task time, was also better in the offspring of mothers who received prenatal FO supplementation. Our results also suggest that the increased maternal tHcy levels predict worse speed processing development at 9 years. These results suggest to devote attention to an adequate maternal LC-PUFAs and folate status during pregnancy.

## Figures and Tables

**Figure 1 nutrients-13-00131-f001:**
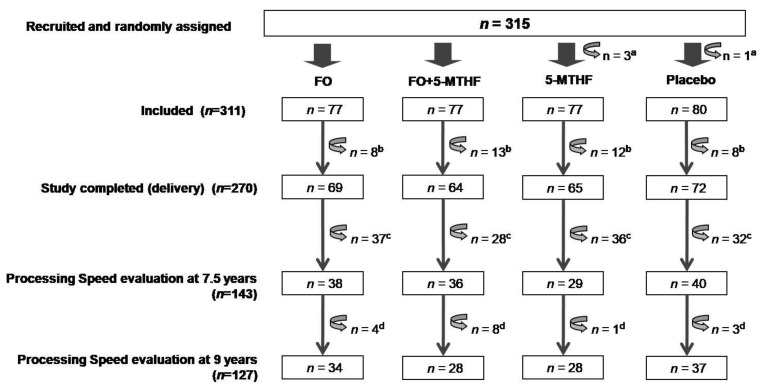
Flowchart of NUHEAL participants up to 9 years. FO: fish-oil; 5-MTHF: 5-methyltetrahydrofolate. ^a^ 4 participants who did not meet the inclusion criteria: 2 women weighed > 92 kg, 1 of whom used commercial FO preparations and 2 women regularly consumed FO preparations. ^b^ 41 participants did not complete the study: noncompliance (*n* = 2), relocation (*n* = 1), aversion to or bad taste of the supplement (*n* = 9), loss of contact (*n* = 2) and unknown reasons (*n* = 27). ^c^ 133 participants lost to follow up at 7.5 years: relocation (*n* = 3), loss of contact (*n* = 74), infants born prematurely (*n* = 4), congenital left-side anophthalmus (*n* = 1), craniosynostosis (*n* = 1), left-side deafness (*n* = 1), unwillingness to continue (*n* = 50). ^d^ Processing speed tasks were not entirely performed in 16 participants at 9 years.

**Figure 2 nutrients-13-00131-f002:**
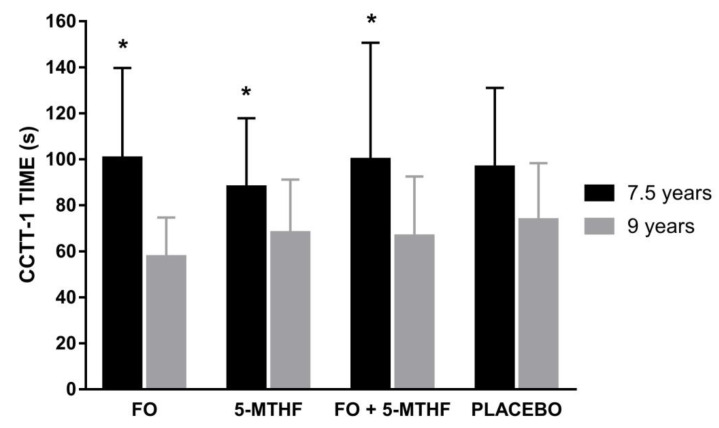
Effects of prenatal supplementation on *CCTT-1* test in children at 7.5 and 9 years. Level of significance was obtained from one-way ANOVA. *** Significant differences between 7.5 years and 9 years were observed in time spent (s) to solve the ***CCTT***-**1** test between children whose mothers were supplemented with FO (*p* = 0.0001), 5-MTHF (*p* = 0.0113) or both treatments (FO + 5MTHF) (*p* = 0.038) during pregnancy. 5-MTHF: 5-Methyltetrahydrofolate; CCTT-1: Children´s Color Trails Test; FO: fish-oil.

**Table 1 nutrients-13-00131-t001:** General characteristics of the studied population.

	FO (*n* = 38)	5-MTHF (*n* = 29)	FO + 5-MTHF (*n* = 36)	Placebo (*n* = 40)	*p*
Maternal age (years)	28.81 ± 5.25	30.70 ± 5.68	29.56 ± 4.32	30.71 ± 3.90	0.231
BMI (kg/m^2^)					
20 weeks	26.03 ± 3.62	24.92 ± 2.45	25.28 ± 2.77	24.74 ± 2.28	0.221
30 weeks	28.52 ± 3.91	26.87 ± 2.43	27.23 ± 2.86	26.91 ± 2.29	0.511
Country of origin					0.102
Spain	28(73.7%)	18(62%)	26(72.2%)	25(62.5%)	
Germany	10(24.3%)	11(38%)	10(27.8%)	15(37.5%)	
Hematocrit (%) at 30 weeks	33.85 ± 3.73	32.61 ± 5.14	33.37 ± 2.58	33.13 ± 2.75	0.662
Parity [*n* (%)]					0.521
0	23 (60.5%)	23 (79.3%)	28 (93.3%)	28 (70%)	
≥1	3 (7.9%)	3 (12.5%)	2 (5.5%)	4 (10%)	
Mode of Delivery					0.220
Spontaneous	22(57.9%)	15(51.7%)	21(58.3%)	18(45%)	
Forceps	2(5.2%)	6(17.2%)	2(5.5%)	2(5%)	
Vacuum	0	0	0	2(5%)	
Cesarean section	4(10.5%)	2(6.9%)	3(8.3%)	6(15%)	
Smoking in pregnancy [Yes = *n* (%)]	7 (18.4%)	5 (17.2%)	7 (19.4%)	4 (10%)	0.251
Gravidity risk at 20 weeks [*n* (%)]					0.652
No risk factors	6 (15.8%)	6 (20.7%)	8 (22.2%)	11 (27.5%)	
≥1 risk factors	18 (47.4%)	18 (62.1%)	22 (61.1%)	21 (52.5%)	
High Maternal education [*n* (%)]	8 (21.05%)	8 (27.58%)	14 (38.89%)	10 (25%)	0.332
Family Status Pregnancy [*n* (%)]					0.124
Single	8 (21.1%)	8 (27.4%)	1 (2.7%)	10 (25%)	
Partnership	23 (60.5%)	23 (44.8%)	29 (80.6%)	29 (72.5%)	
Gestational Age (weeks)	38.90 ± 1.51	38.75 ± 1.62	38.73 ± 2.06	39.43 ± 1.43	0.622
AA/DHA ^1^	0.90 ± 0.38	1.13 ± 0.54	0.97 ± 0.42	1.14 ± 0.63	0.098
Plasma Folate (µg/L) ^1^	6.17 ± 4.33	12.10 ± 5.55	13.46 ± 5.78	6.06 ± 0.82	**<0.001**
tHcy (µmol/L) ^1^	7.08 ± 2.83	6.29 ± 2.81	6.93 ± 3.09	6.78 ± 2.39	0.952

Data are presented as *n* (%) for categorical data, and mean ± SDs for parametrically distributed data.^1^: Values obtained from mothers at delivery. *p*: level of significance from one-way ANOVA for continuous variables or χ^2^ test for categorical variables. Bold: *p* value < 0.05. Noted that there are missing values for some descriptive variables. 5-MTHF: 5-methyltetrahydrofolate; AA: arachidonic acid; DHA: docosahexaenoic acid; FO: fish-oil; tHcy: total homocysteine. *n* = number of cases.

**Table 2 nutrients-13-00131-t002:** Descriptive analysis of processing speed evaluation in the NUHEAL children at 7.5 and 9 years.

	FO		5-MTHF		FO + 5-MTHF		Placebo		*p*
Children at 7.5 Years	*n*	Mean ± SD	*n*	Mean ± SD	*n*	Mean ± SD	*n*	Mean ± SD	
SDMT—hits	36	23.56 ± 7.62	28	22.61 ± 6.87	34	22.38 ± 5.27	41	24.66 ± 6.38	0.204
CCTT-1 (sc)	36	100.44 ± 39.27	28	87.85 ± 30.05	34	96.56 ± 34.48	41	99.88 ± 50.77	**0.017**
STROOP Test—hits-1 (Word-reading)	37	59.49 ± 17.72	28	61.82 ± 11.66	34	57.68 ± 16.34	41	59.34 ± 16.41	0.155
STROOP Test—hits-2 (Color-naming)	37	38.97 ± 7.77	28	42.14 ± 7.26	34	42.18 ± 9.36	41	41.95 ± 8.41	0.523
**Children at 9 years**									
SDMT—hits	33	31.42 ± 6.66	26	30.00 ± 7.16	28	30.25 ± 7.73	37	33.76 ± 8.60	0.500
CCTT-1 (sc)	33	57.60 ± 17.13	28	67.99 ± 23.16	28	73.61 ± 24.73	37	66.58 ± 25.99	0.112
STROOP Test—hits-1 (Word-Reading)	34	70.53 ± 12.69	28	71.54 ± 10.29	28	72.32 ± 8.71	37	73.62 ± 9.65	0.187
STROOP Test—hits-2 (Color-Naming)	34	48.26 ± 7.58	28	49.32 ± 9.49	28	50.61 ± 8.57	37	51.41 ± 7.94	0.627

*p*: level of significance obtained from one-way ANOVA test; Bold: *p* < 0.05. 5-MTHF: 5-methyltetrahydrofolate; CCTT-1: Children Color Trails Test; FO: fish-oil; SDMT: Symbol Digit Modalities Test; *n* = number of cases.

**Table 3 nutrients-13-00131-t003:** Effect of selected cofounders on the processing speed outcomes in the NUHEAL children. Only significant associations are shown.

	Categories	*b*	CI (95%)		*p*
			LCL	UCL	
**Symbol Digit Modalities Test (SDMT) (Hits)**					
Time point	7.5 years	0	-	-	-
	9 years	10.72	7.98	13.47	<0.001
dBMI (kg/m^2^)		−0.63	−1.25	−0.02	0.043
AA/DHAratio ^1,2^		8.80	3.61	13.99	0.001
tHcy (µmol/L) ^1^		−0.52	−1.02	−0.002	0.049
**Children Color Trails Test (CCTT−1) (sc)**					
Time point	7.5 years	0	-	-	-
	9 years	−34.12	−46.92	−21.10	<0.001
Smoking	No	0	-	-	-
	Yes	23.43	9.43	37.44	0.001
Mode of delivery	Spontaneous	0	-	-	-
	Forceps	−5.91	−25.07	13.24	0.545
	Vacuum	24.77	6.01	43.53	0.010
	Cesarean section	2.66	−9.15	14.47	0.659
AA/DHAratio ^1,2^		−27.87	−51.13	−4.61	0.019
**Word-Reading Stroop Test (WRST) (hits)**					
Time point	7.5 years	0	-	-	-
	9 years	15.37	11.20	19.55	<0.001
Sex	Girls	0	-	-	-
	Boys	−4.93	−9.06	−0.81	0.019
tHcy (µmol/L) ^1^		−1.09	−2.04	−0.14	0.025
**Color-Naming Stroop Test (CNST) (hits)**					
Time point	7.5 years				
	9 years	10.00	6.80	13.21	**<0.001**
AA/DHA ratio ^1,2^		6.55	0.50	12.60	**0.034**

^1^: values obtained from mothers at delivery; ^2^: fatty acids measured as % of total fatty acids. *b*: Regression coefficient; CI: Confidence Interval; LCL: lower confidence limit; UCL; upper confidence limit; *p:* level of significance obtained from mixed model analysis; dBMI: difference of body mass index between 20 and 30 weeks of pregnancy; AA: Arachidonic acid; DHA: Docosahexaenoic acid; tHcy: total homocysteine. Bold: *p* value < 0.05.

**Table 4 nutrients-13-00131-t004:** Logistic regression analysis after dichotomizing children aged 9 years of age below or above P75, P50 or P25 for Symbol Digit Modalities Test (***SDMT***), Children Color Trails Test ***(CCTT-1)*** and ***STROOP*** Test. Only significant associations are shown.

	Maternal AA/DHA	*FADS1* rs174556		Maternal tHcy	Maternal High Education	Maternal Age
		1	2			
***SDMT*—hits P75** Contrast 95% CI *p*-value	30.46 (3.68, 252.0) **0.002**	3.31 (1.28, 8.56) **0.014**	1.43 (0.31, 6.55) 0.64	0.71 (0.56, 0.89) **0.003**	3.63 (1.32, 9.98) **0.013**	1.04 (0.94, 1.14) 0.446
***CCTT-1* (sc) P25** Contrast 95% CI *p*-value	18.53 (2.13, 160.9) **0.008**	2.75 (1.02, 7.39) **0.045**	4.95 (0.91, 27.07) 0.065	1.03 (0.85, 1.26) 0.733	2.03 (0.71, 5.77) 0.184	1.03 (0.93, 1.14) 0.587
***STROOP* Test—hits 1** ***(Word-reading)* P50** Contrast 95% CI *p*-value	3.73 (0.31, 44.28) 0.297	0.63 (0.19, 2.03) 0.439	1.67 (0.22, 12.85) 0.624	0.69 (0.51, 0.93) **0.015**	1.47 (0.44, 4.85) 0.529	0.95 (0.84, 1.08) 0.440
***STROOP* Test—hits 2** ***(Color-naming)* P75** Contrast 95% CI *p*-value	21.70 (0.88, 534.97) 0.06	0.28 (0.07, 1.16) 0.08	0.67 (0.07, 6.48) 0.733	0.51 (0.32, 0.81) **0.005**	0.69 (0.17, 2.76) 0.602	1.20 (1.01, 1.42) **0.039**

*p*: level of significance obtained from logistic regression mixed model; Bold: *p* < 0.05. 1 = Heterozygous, 2 = Homozygous minor; AA: arachidonic acid; CCTT-1: Children Color Trails Test; DHA: Docosahexaenoic acid; *FADS*: Fatty Acid Desaturase; SDMT: Symbol Digit Modalities Test; tHcy: total homocysteine.

## Data Availability

Not applicable.

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
