# Peer review of "Long-Chain Polyunsaturated Fatty Acids, Homocysteine at Birth and Fatty Acid Desaturase Gene Cluster Polymorphisms Are Associated with Children’s Processing Speed up to Age 9 Years"

_nutrients, 2020, doi:10.3390/nu13010131_

Round 1

Reviewer 1 Report

This paper discusses, for the first time, the effects of supplements and genetic makeup on processing time in children up to 9 year old. It is well written, the literature review and analyses is extensive and thorough and the methods are sound. The paper contributes to a better understanding of the effects of the aforementioned variables on neuropsychological domains.

Author Response

Thank you dear Reviewer for your kind words,

Best regards

Cristina Campoy

Reviewer 2 Report

I have read the manuscript: ‘Long-chain polyunsaturated fatty acids, homocysteine at birth and fatty acid desaturase gene cluster polymorphisms are associated with children´s processing speed up to age 9 years.’ The authors describe a follow-up study of children whose mothers were supplemented with fish oil, fish oil + folic acid, folic acid, or placebo during pregnancy. Children were followed-up numerous time-points up to age 9.5 years. The results reported in the current manuscript are focused on the information processing speeds measured at age 7.5 and 9 years. The manuscript is well written, I do of course have some comments please see my comments below.

I am not sure what the rationale is to do the processing speed analyses at both age 7.5 and 9 years. Would you expect that prenatal supplementation leads to a different effect on age 7.5 years than age 9 years? Why did you use 7.5 years as reference variable in your analyses (for example table 4)?

The ClinicalTrials registry indicates that neurocognitive measures were collected at multiple time points ( age 4 years, 5.5 years, 6.5 years, 7.5 years, 8 years, 9 years, 9.5 years) were the processing speed measures collected at all time points or selected time points? If processing speed measures were collected a multiple time points, why did you specifically chose to report the results for age 7.5 and 9?

Line 73: The speed of information processing is essential for higher order cognitive functions, including  memory or executive functions [1]. These cognitive processes can be defined as the time required to move information from one neuron to the next.
Should this read ‘Information processing can be defined’?

Line 82: Interestingly, their effects on neurodevelopment depend of the timing of occurrence and brain needs for particular nutrients at
Should ‘of’ be ‘on’?

Line 113: as well as maternal FADS polymorphisms
Is there a specific reason as to why you measured the FADS polymorphism in the mothers and not the child?

Line 129: Processing speed tasks were entirely performed in 143 NUHEAL children,
Should you specify that processing speed tasks were entirely performed in 143 children at age 7.5? Fewer children were included at age 9 according to figure 1.

Line 147: NUTRIMENTHE Neuropsychological Battery (NNB) was used to evaluate the whole spectrum of neuropsychological functioning in children aged 7.5-9 years, including processing speed tasks
Were the children tested once or twice with the NNB?

Line 171: Both word-reading and color-naming scores are related to processing speed,
Do word-reading and color-naming measure other aspects of processing speed or the same?

Statistical analyses: The number of statistical tests executed is very large. You describe both a multiple linear regression model and a mixed model analyses; it is unclear why you would do both analyses. Did you analyses other outcome variables? Additionally it unclear how the mixed model were built? How were the covariates/confounders added (one by one forward? All at ones?)

Line 224: including maternal age, maternal education level, maternal smoking, maternal BMI, mode of delivery, gestational age and child´s sex

Why did you include these covariables?

Line 231: Logistic regression mixed model was used when dependent variables were dichotomized (below or above third quartile). Why would you use dichotomized variables when you already have continuous variables for Symbol Digit Modalities Test for example. Additionally, why were these data dichotomized at below and above the 3th quartile? Lastly, in table 6 it is noted that children were divided below 25, below, 50 and below 75 which would suggest you have three group (i.e. not dichotomized as that would indicate two groups). What did you do with the data of children that scored above 75? Please also note that the tables are not in order in which they are mentioned in the text.

On line 234 you mention: ‘Interactions between groups of prenatal supplementation and confounders were analyzed in each multivariate analysis performed, but significant differences were not observed’. While in table 3 study group * time point interaction is included.  Additionally, this seems to be a results, which should not be reported in the methods but in the results. Lastly, table 5 and 6 in the supplementary files do not seem to contain results of the suggested analyses.

Line 238: P values < 0.05 were considered as statistically significant
Considering the large amount of statistical test you have conducted, wouldn’t it be prudent to use an adapted p?

Table 1: Mode of delivery is missing. Why do you include AA/DHA ratio while you supplemented with DHA + EPA, wouldn’t it be more logical to report the DHA and EPA levels? Could you please specify the P-values in this table instead of NS?
If the supplementation did not have an influence on AA/DHA ratio at delivery can you be certain that the mothers consumed your supplements during the pregnancy? Considering they took 500mg DHA and 150mg EPA per day for approximately 20wks, wouldn’t you expect an increase in their levels?

Line 265: In fact, children born to mothers who were supplemented with 5-MTHF during pregnancy showed a decrease in the timing to solve
It would be handy for your readers if you mention (somewhere) whether an increase/decrease in time on each of the tests is a positive or negative outcome.

Line 285: after adjusting for maternal blood parameters, mother´s genetic polymorphisms and lifestyle 285 variables in pregnancy
In the method you do not specify that the analyses will be adjusted for maternal blood parameters (which specifically), polymorphism or lifestyle variables (which specifically?). Please specify this in the methods.

Line 322: supplementation, we observed that decrease in time elapsed was higher in those children whose mothers were supplemented with FO during pregnancy (p = 0.0001), 5-MTHF (p = 0.0113) or FO+5-MTHF (p = 0.038) (Figure 2)

The text in the main text is not the same as the text under the graph: Significant differences at 7.5 years and 9 years were observed in time spent (s) to solve the CCTT- test between children whose mothers were supplemented with FO (p = 0.0001

It would be strange if you did not find an difference between the ages 7.5 and 9 years, children develop and thus improve their processing speed. What is it that you want to tell with this text/figure?

Line 334: As shown in Table 4, other selected cofounders had also a significant effect on the time required 334 to complete CCTT-1 test

As you did not change the confounders in the mothers, this is not an effect (i.e. causal), but an association/relationship. Same is true for other sentences in the manuscript.

Table 6: Why are not all tertiles mentioned? i.e SDMT P 25 and P50 are not mentioned.

Line 426: our secondary exploratory analyses seem to indicate that maternal AA/DHA ratio and FADS1 rs174556 SNPs, were positively associated with later processing speed in their offspring up to 9 years, while tHcy concentrations in maternal plasma at delivery showed a negatively effect on child processing speed\

Please specify that these findings relate to specific measures of processing speed and not all measures of processing speed you utilized.

Line 473: our results show positive and strong long-term effects of 473 perinatal LC-PUFAs, in terms of adequate AA/DHA ratio and FADS1 polymorphism, on cognitive 474 development,

Considering the large amount of analyses you executed and the limited number of significant findings, I feel this is a rather far-fetched conclusion.

Line 480: Given that country of origin has been accounted for as a confounder

This is not mentioned in the methods.

Author Response

Response to Reviewer 2 Comments

I have read the manuscript: ‘Long-chain polyunsaturated fatty acids, homocysteine at birth and fatty acid desaturase gene cluster polymorphisms are associated with children´s processing speed up to age 9 years.’ The authors describe a follow-up study of children whose mothers were supplemented with fish oil, fish oil + folic acid, folic acid, or placebo during pregnancy. Children were followed-up numerous time-points up to age 9.5 years. The results reported in the current manuscript are focused on the information processing speeds measured at age 7.5 and 9 years. The manuscript is well written, I do of course have some comments please see my comments below.

We really appreciate the comments arisen from the Reviewer and we are thankful for them as they would improve for sure our manuscript. Point-to-point answers follow your questions and comments below.

We used the "Track Changes" function in Microsoft Word for the revision.  

I am not sure what the rationale is to do the processing speed analyses at both age 7.5 and 9 years. Would you expect that prenatal supplementation leads to a different effect on age 7.5 years than age 9 years? Why did you use 7.5 years as reference variable in your analyses (for example table 4)?

Response: Thank you very much for your comment. 

This project was aimed to evaluate the effects of nutrition during the early stages of life on child neurodevelopment and cognitive performance. Children included in current study were programmed to be examined using NNB (NUTRIMENTHE Neuropsychological Battery-NNB) twice within NUTRIMENTHE framework (funding available for such purpose), when they were aged 7.5 and 9 years.

We hypothesized that maternal supplementation during pregnancy would have long-term effects at 7.5 years, and having a second time at 9 years old, we wanted to ensure that the effect remains in time (double-checking). For this purpose, mixed models were performed to also study the time effect (7.5 years as reference), to discard its potential influence on the results.

The ClinicalTrials registry indicates that neurocognitive measures were collected at multiple time points ( age 4 years, 5.5 years, 6.5 years, 7.5 years, 8 years, 9 years, 9.5 years) were the processing speed measures collected at all time points or selected time points? If processing speed measures were collected multiple time points, why did you specifically chose to report the results for age 7.5 and 9?

Response: We thank the reviewer for this comment.

Indeed, NUHEAL is a long-term follow-up study, where different non-invasive and age adequate neurocognitive measures were collected in the children at different time-points. At 4, 5.5 and 6.5 years processing speed was not specifically measured. The design of the NBB permit us to study the brain development by specific domains.  Then, as current study was framed within NUTRIMENTHE Project, the ages where processing speed function was specifically measured in NUHEAL children were at 7.5 and 9 years old.

This information has been now included in manuscript: “Within NUTRIMENTHE EU Project (grant agreement no. 212652) framework,…”

Line 73: The speed of information processing is essential for higher order cognitive functions, including memory or executive functions [1]. These cognitive processes can be defined as the time required to move information from one neuron to the next.
Should this read ‘Information processing can be defined’?

Response: We agree with the Reviewer that this expression is incorrect. Therefore, it has been modified as "Processing speed can be defined as the time required to move- information from one neuron to the next". Consequently, “This cognitive process is highly related intact myelination,….” , have been also modified.

Line 82: Interestingly, their effects on neurodevelopment depend of the timing of occurrence and brain needs for particular nutrients at
Should ‘of’ be ‘on’?

Response: Thank you for your observation.

We have replaced "depend of the timing" for "depend on timing".

Line 113: as well as maternal FADS polymorphisms
Is there a specific reason as to why you measured the FADS polymorphism in the mothers and not the child?

Response: We appreciate the thoughtful Reviewer's comment.

According to our primary aim, we only focused on maternal FADS polymorphisms in order to understand how nutritional intervention during pregnancy may affect later processing speed in the offspring. In this regard, as noted in both Introduction and Discussion sections, the fetus has a very limited capacity for PUFAs elongation and desaturation, and its availability is suggested to be influenced by maternal FADS polymorphisms (Steer, C.D.; Lattka, E.; Koletzko, B.; Golding, J.; Hibbeln, J.R. Maternal fatty acids in pregnancy, FADS polymorphisms, and child intelligence quotient at 8 y of age. Am. J. Clin. Nutr 2013, 98, 1575-1582, doi: 10.3945/ajcn.112.051524.) having long-term effects on the offspring's neurodevelopment. From our point of view, these analyses of variants in FADS genes provide additional insights into the role of FAs during a very critical-time frame for brain development and the potential long-term effects later in life.

Line 129: Processing speed tasks were entirely performed in 143 NUHEAL children,
Should you specify that processing speed tasks were entirely performed in 143 children at age 7.5? Fewer children were included at age 9 according to figure 1.

Response: We agree with the Reviewer and acknowledge his/her kind observation. In order to clarify this issue, and according to data shown in Figure 1, sample sizes of processing speed evaluation at both 7.5 and 9 years old ages have now been included, as follows:

"Processing speed tasks were entirely performed in 143 NUHEAL children at 7.5 years (FO, n=38; 5-MTHF, n=29; FO+5-MTHF, n=36; placebo n=40) and 127 NUHEAL children at 9 years (FO, n=34; 5-MTHF, n=28; FO+5-MTHF=28; placebo, n=37) (Figure 1)”.

This information has been also included in the Abstract.

Line 147: NUTRIMENTHE Neuropsychological Battery (NNB) was used to evaluate the whole spectrum of neuropsychological functioning in children aged 7.5-9 years, including processing speed tasks.
Were the children tested once or twice with the NNB?

Response: We appreciate the Reviewer comment. We agree with Reviewer that the sentence as it stood is misleading and inappropriate, and we would like to personally apologize for this discrepancy. Evaluation of whole spectrum of neuropsychological functioning using NUTRIMENTHE Neuropsychological Battery (NNB) was carried out once in NUHEAL children at 7.5 years old and again at 9 years.

In order to clarify this issue, following change has been performed: "NUTRIMENTHE Neuropsychological Battery (NNB) was used to evaluate the whole spectrum of neuropsychological functioning in children aged 7.5 and 9 years, ..."

Line 171: Both word-reading and color-naming scores are related to processing speed,
Do word-reading and color-naming measure other aspects of processing speed or the same?

Response: We really appreciate the comments from the Reviewer. In the used version of SCWT, both word-reading and color-naming scores represent the "congruous condition" in which participants are required to read names of colors (word-reading) printed in black ink and name different color patches (color-naming), as stated in methods (lines 166-169). Consequently, both scores were used to assess the same aspect of processing speed, that is, they asses the processing speed of congruent semantic information.

Such clarification has been directly included in the manuscript, lines 176-178: "Both word-reading and color-naming scores (hits) are related to processing speed of congruent semantic information (high scores represent better processing speed), but..."

Statistical analyses: The number of statistical tests executed is very large. You describe both a multiple linear regression model and a mixed model analyses; it is unclear why you would do both analyses. Did you analyses other outcome variables? Additionally it unclear how the mixed model were built? How were the covariates/confounders added (one by one forward? All at ones?)

Response: We agree with the reviewer; the paragraph was misleading. We only did mixed models.The text has been rewritten to clearly state the methods. Our outcomes rely on the processing speed tests: SDMT, CCTT-1 and SCWT 1 and 2. These are continuous outcomes, at two different time points, for which we used multilevel models, where confounders were added all at once. Furthermore, in psychology is quite common to percentile the outcomes to assess the performance according to population percentiles. In this case, for a categorical outcome, we implemented logistic regression mixed models. We have rewritten the paragraph to add more insight.

“Variables of processing speed obtained from SDMT, CCTT-1 and SCWT tests were considered as dependent variables. Considering the hierarchical structure resulted from processing speed evaluation in children at 7.5 and 9 years, mixed models for repeated measures were performed considering as fixed effects a set of potential confounding variables, such as study group, time point, country of origin, maternal age, hematocrit levels,  parity, gravidity risk maternal education level, maternal smoking, maternal BMI, mode of delivery, gestational age, child´s sex and mother’s biochemical and molecular parameters at delivery (AA/DHA ratio, FADS1 rs174556 polymorphism, total homocysteine , MTHFR C677T polymorphisms, plasma folate). Interactions between study group and time point were also studied. The random effects were the subjects nested ID to estimate the intra variance. Logistic regression mixed model was used when dependent variables were dichotomized (below or above percentile 25, 50 and 75), in which case the measure of the effect was the suitable odds ratio with corresponding confidence interval. All potential confounders were added at once and selected those which significance p<0.05.”

Line 224: including maternal age, maternal education level, maternal smoking, maternal BMI, mode of delivery, gestational age and child´s sex

Why did you include these covariables?

Response: Thank you so much for your interesting comment. We have included these confounders and co-variables due to their associations with neuropsychological performance in children, as we have previously reported in the following studies: Escribano, J.; Luque, V.; Canals-Sans, J.; Ferre, N.; Koletzko, B.; Grote, V.; Weber, M.; Gruszfeld, D.; Szott, K.; Verduci, E., et al. Mental performance in 8-year-old children fed reduced protein content formula during the 1st year of life: safety analysis of a randomised clinical trial. Br. J. Nutr 2016, 2, 22-30,  and Pérez-García, M., de Dios Luna, J., Torres-Espínola, F. J., Martínez-Zaldívar, C., Anjos, T., Steenweg-de Graaff, J., et al. (2019). Cultural effects on neurodevelopmental testing in children from six European countries: an analysis of NUTRIMENTHE Global Database. Br. J. Nutr 2019, 122(s1), S59-S67.

Statistical methods have been re-written to clarify this issue as stated above.

Line 231: Logistic regression mixed model was used when dependent variables were dichotomized (below or above third quartile). Why would you use dichotomized variables when you already have continuous variables for Symbol Digit Modalities Test for example. Additionally, why were these data dichotomized at below and above the 3th quartile? Lastly, in table 6 it is noted that children were divided below 25, below, 50 and below 75 which would suggest you have three group (i.e. not dichotomized as that would indicate two groups). What did you do with the data of children that scored above 75? Please also note that the tables are not in order in which they are mentioned in the text.

Response: Thank you so much for your observation. In psychology, it is quite common to percentile the outcomes in order to assess the performance according to population percentiles. Data were dichotomized at p25, p50 and p75, and only the significant associations were shown in Table 6. For each outcome we have only two groups. We wanted to assess those children who performed “better” or “worse” according to percentile populations.

Statistical methods have been rewritten to clarify this issue as stated above.

On line 234 you mention: ‘Interactions between groups of prenatal supplementation and confounders were analyzed in each multivariate analysis performed, but significant differences were not observed’. While in table 3 study group * time point interaction is included.  Additionally, this seems to be a results, which should not be reported in the methods but in the results. Lastly, table 5 and 6 in the supplementary files do not seem to contain results of the suggested analyses.

Response: We agree with the reviewer; we have combined tables 3, 4 and 5 in a single table (table 3), showing only the significant confounders for each processing speed outcome (SDMT, CCTT-1 and SCWT).

Statistical methods have been re-written to clarify this issue as stated above.

Line 238: values<0.05 were considered as statistically significant
Considering the large amount of statistical test you have conducted, wouldn’t it be prudent to use an adapted p?

Response: We apologize for the misunderstanding. The statistical methods have been re-written and we believe with this statement, choosing a p-value< 0.05 is prudent. In fact, post-hoc comparison has been made to ensure the quality of our results.

Table 1: Mode of delivery is missing. Why do you include AA/DHA ratio while you supplemented with DHA + EPA, wouldn’t it be more logical to report the DHA and EPA levels? Could you please specify the P-values in this table instead of NS?
If the supplementation did not have an influence on AA/DHA ratio at delivery can you be certain that the mothers consumed your supplements during the pregnancy? Considering they took 500mg DHA and 150mg EPA per day for approximately 20wks, wouldn’t you expect an increase in their levels?

Response: Thank you so much for your interesting comment.

Mode of delivery and p-values have been included in table 1.

In our previous NUHEAL publication, it has been already published the effects of prenatal supplementation with fish oil on DHA, EPA and AA levels; in fact, mothers supplemented with FO showed an increase of DHA and EPA levels. However, despite such supplementations, no effects were observed in the means values of AA levels(Krauss-Etschmann, S.; Shadid, R.; Campoy, C.; Hoster, E.; Demmelmair, H.; Jiménez, M.; Gil, A.; Rivero, M.; Veszpremi, B.; Decsi, T., et al. Effects of fish-oil and folate supplementation of pregnant women on maternal and fetal plasma concentrations of docosahexaenoic acid and eicosapentaenoic acid: a European randomized multicenter trial. Am. J. Clin. Nutr 2007, 85, 1392-1400, doi: 10.1093/ajcn/85.5.1392.). Thus, AA/DHA ratio was considered better biomarker reflecting the proportions of these two important long-chain polyunsaturated fatty acids and their equilibrium, considering that maternal LC-PUFAs nutritional intake differed between subjects; nevertherless, 90% of the NUHEAL pregnant women achieved the recommended dietary intakes for DHA, as we have previously published (Verwied-Jorky S, Campoy C, Trak-Fellermeier M, Decsi T, Dolz V, Koletzko B. Dietary intake of natural sources of docosahexaenoic acid and folate in pregnant women of three European cohorts. Ann Nutr Metab. 2008;53(3-4):167-74. ).  During the follow up of the pregnant mothers, we controlled the amount of intake of the product. Subjects were instructed to return leftover sachets to the study center. Compliance was assessed in standardized questionnaires at gestation week 30 and at delivery by asking each subject how many days of dosing she had missed (eg, 6, 5, or none).

Line 265: In fact, children born to mothers who were supplemented with 5-MTHF during pregnancy showed a decrease in the timing to solve
It would be handy for your readers if you mention (somewhere) whether an increase/decrease in time on each of the tests is a positive or negative outcome.

Response: We apologize for the lack of information on the method used and its interpretation. In order to solvent this issue, we have rewritten the neuropsychological assessment section as follows:

 “The number of correctly identified symbols (hits) is recorded as a score, interpreting higher scores to be an indicator of better child processing speed”.

 “The final score is the time (in seconds) taken to complete part-1 of the CCTT [35], so that shortest time in this task is related to better processing speed of participant”.

 “Both word-reading and color-naming scores (hits) are related to processing speed of congruent semantic information (high scores represent better processing speed)”

Line 285: after adjusting for maternal blood parameters, mother´s genetic polymorphisms and lifestyle variables in pregnancy
In the method you do not specify that the analyses will be adjusted for maternal blood parameters (which specifically), polymorphism or lifestyle variables (which specifically?). Please specify this in the methods.

Response: We would like to thank you for your comment. We adjusted our analysis including all variables at once, and selected those variables that showed any significance with the outcome variable. This information has now been correct in manuscript.

Line 322: supplementation, we observed that decrease in time elapsed was higher in those children at 9 years old whose mothers were supplemented with FO during pregnancy (= 0.0001), 5-MTHF (p = 0.0113) or FO+5-MTHF (p = 0.038) (Figure 2), compared to children at 7.5 years old.

The text in the main text is not the same as the text under the graph: Significant differences at 7.5 years and 9 years were observed in time spent (s) to solve the CCTT-1 test between children whose mothers were supplemented with FO (= 0.0001

It would be strange if you did not find an difference between the ages 7.5 and 9 years, children develop and thus improve their processing speed. What is it that you want to tell with this text/figure?

Response: We fully agree with the reviewer and we apologize for the mistake. Text has been corrected. There are differences between 7.5 and 9 years regarding processing speed in all supplementation groups (*) except the Placebo group. We can appreciate that children at 9 years are able to perform the CCTT-1 task in less time than children at 7.5 years.

Figure caption has been changed.

Line 334: As shown in Table 4, other selected cofounders had also a significant effect on the time required 334 to complete CCTT-1 test

As you did not change the confounders in the mothers, this is not an effect (i.e. causal), but an association/relationship. Same is true for other sentences in the manuscript.

Response: We really appreciate the Reviewer´s observation and, according to his/her suggestion, we have changed the terminology used (association instead of effect)

Table 6: Why are not all tertiles mentioned? i.e SDMT P 25 and P50 are not mentioned.

Response: we apologize again. The 4 selected outcomes were percentilized three times, Above/below p25, p50 and p75. In this table we included only the significant associations from all the dichotomization made above.

Line 426: our secondary exploratory analyses seem to indicate that maternal AA/DHA ratio and FADS1 rs174556 SNPs, were positively associated with later processing speed in their offspring up to 9 years, while tHcy concentrations in maternal plasma at delivery showed a negatively effect on child processing speed\

Please specify that these findings relate to specific measures of processing speed and not all measures of processing speed you utilized.

Response: Thank you very much for your suggestion. Description of these results have been corrected and modified in the manuscript to clarify it as follows:

 “However, our secondary exploratory analyses seem to indicate that maternal AA/DHA ratio and FADS1 rs174556 SNPs, were positively associated with later processing speed in their offspring up to 9 years, particularly in SDMT and CCTT-1 tests, while tHcy concentrations in maternal plasma at delivery showed a negative effect on child processing speed, according to results obtained in SDMT, WRST and CNST tests.”

Line 473: our results show positive and strong long-term effects of 473 perinatal LC-PUFAs, in terms of adequate AA/DHA ratio and FADS1 polymorphism, on cognitive 474 development,

Considering the large amount of analyses you executed and the limited number of significant findings, I feel this is a rather far-fetched conclusion.

Response: We fully agree with the Reviewer that the expression is not correct. Therefore, it has been modified as follows: “our results seem to show positive and strong long-term effects of perinatal LC-PUFAs, in terms of adequate AA/DHA ratio and FADS1 polymorphism, on cognitive development”

Line 480: Given that country of origin has been accounted for as a confounder

This is not mentioned in the methods.

Response: We appreciate the Reviewer´s observation. Country of origin has been added in table 1. Children from Hungary were not included in the present analysis due to a very high number of missing values.

NOTE:

Tables 3, 4 and 5 became Table 3.

Table 6 is now Table 4 due to re-organizations.

Thank you so much for your comments; we really appreciated the time and efforts that you have dedicated to our manuscript.